# A novel domain within the CIL regulates egress of IFITM3 from the Golgi and reveals a regulatory role of IFITM3 on the secretory pathway

Li Zhong[1], Yuxin Song[1], Federico Marziali[1], Rustem Uzbekov[2,3] (ID), Xuan-Nhi Nguyen[1], Chloé Journo[1] (ID), Philippe Roingeard[2,4] (ID), Andrea Cimarelli[1] (ID)

The InterFeron-Induced TransMembrane proteins (IFITMs) are members of the dispanin/CD225 family that act as broad viral inhibitors by preventing viral-to-cellular membrane fusion. In this study, we uncover egress from the Golgi as an important step in the biology of IFITM3 by identifying the domain that regulates this process and that similarly controls the egress of the dispanins IFITM1 and PRRT2, protein linked to paroxysmal kinesigenic dyskinesia. In the case of IFITM3, high levels of expression of *wild-type*, or mutations in the Golgi egress domain, lead to accumulation of IFITM3 in the Golgi and drive generalized glycoprotein trafficking defects. These defects can be relieved upon incubation with Amphotericin B, compound known to relieve IFITM-driven membrane fusion defects, as well as by v-SNARE overexpression, suggesting that IFITM3 interferes with membrane fusion processes important for Golgi functionalities. The comparison of glycoprotein trafficking in WT versus IFITMs-KO cells indicates that the modulation of the secretory pathway is a novel feature of IFITM proteins. Overall, our study defines a novel domain that regulates the egress of several dispanin/CD225 members from the Golgi and identifies a novel modulatory function for IFITM3.

## Introduction

The InterFeron-Induced TransMembrane proteins (IFITMs) are broad viral inhibitors that prevent membrane fusion between viral and cellular membranes, thus protecting the cell from infection (1). IFITMs belong to the dispanin/CD225 family of proteins that originated from metazoan lineages and diversified into four distinct subfamilies (A through D) that include proteins of known and yet unknown functions (2). The A subfamily regroups all IFITMs which in humans are IFITM1, 2, and 3, which we will refer to hereafter as IFITMs, that are IFN-induced proteins essentially studied in the context of viral infection (3); IFITM5 that plays undefined roles in Osteogenesis imperfecta type V, a bone-specific disease (4) and IFITM10 whose functions are unknown. An additional known member of the dispanin/CD225 family is the neuron specific PRoline-Rich Transmembrane protein 2 (PRRT2, B subfamily) that is involved in neurotransmitter vesicles regulation and has been genetically linked to benign familial infantile seizures and to paroxysmal kinesigenic dyskinesia (PKD), the most common type of paroxysmal movement disorder (5).

Members of the dispanin/CD225 family are characterized by a similar structure consisting of an intramembrane domain (IMD, previously defined as transmembrane domain 1, TM1), a cytoplasmic intracellular loop (CIL) and a transmembrane domain TMD (previously defined as transmembrane domain 2, TM2, as schematically shown in Fig 1A for IFITM3). By virtue of this structure, IFITMs decorate cellular membranes in which they act as membrane fusion inhibitors. The distribution and activities of IFITMs on cellular membranes are highly regulated and are driven by distinct N and C termini and also by several post-translational modifications and in particular palmitoylation on cysteine residues, ubiquitination and methylation on lysines, as well as phosphorylation on a specific tyrosine residue (6, 7, 8, 9, 10). Overall, these changes and domains concur in the more prominent distribution of IFITM1 at the plasma membrane and of IFITM2 and 3 at intracellular ones. More recently, IFITMs have been linked to cellular functions independent from their protective role against viral infection in the regulation of glucose metabolism in mice, of phosphatidylinositol 3-kinase (PI3K)–mediated signaling in B cells (11) and of trophoblast fusion during placental formation in vivo (12), overall supporting the notion that IFITM proteins likely act well beyond a viral context.

In this study, through the extensive characterization of an IFITM3 CIL mutant, we have identified a novel protein domain that regulates the normal egress of IFITM3 from the Golgi apparatus. This

---

[1]Centre International de Recherche en Infectiologie (CIRI), Univ Lyon, Inserm, U1111, Université Claude Bernard Lyon 1, CNRS, UMR5308, ENS de Lyon, Lyon, France [2]Plateforme IBiSA de Microscopie Electronique, Université de Tours et CHU de Tours, Tours, France [3]Faculty of Bioengineering and Bioinformatics, Moscow State University, Moscow, Russia [4]INSERM U1259, Université de Tours et CHU de Tours, Tours, France

Correspondence: acimarel@ens-lyon.fr

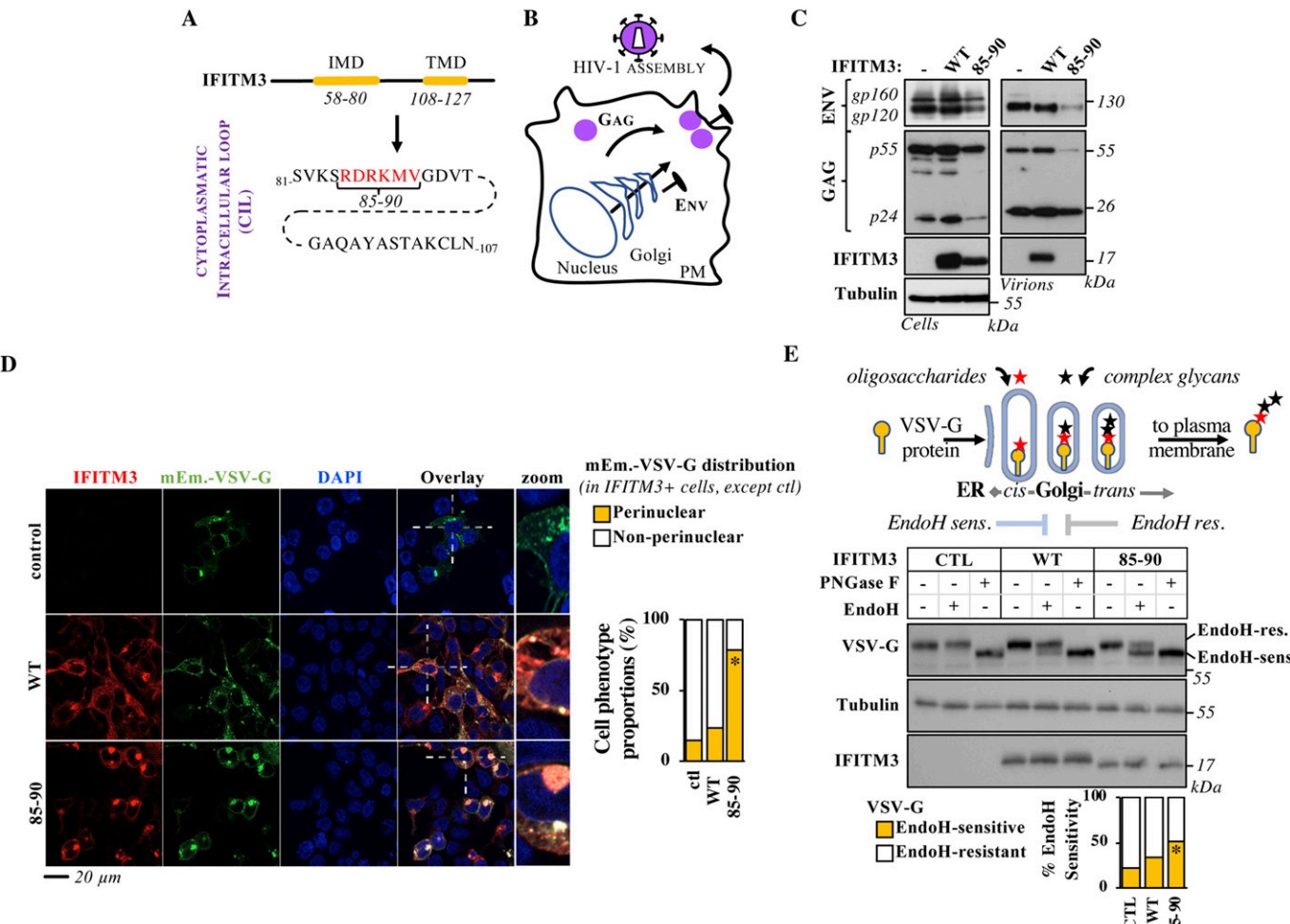

**Figure 1. A novel gain-of-function IFITM3 mutant induces secretory pathway defects that affect glycoproteins trafficking.**
**(A)** Genomic structure of IFITM3 with a highlight on amino acids of the cytoplasmic intracellular loop. In mutant 85-90, red residues have been mutated to alanine.
**(B)** Schematic representation of HIV-1 virion assembly. PM stands for plasma membrane. **(C)** HEK293T cells were transfected with DNAs coding for the HIV-1 proviral clone NL4-3 along with control, WT or 85-90 mutant IFITM3. Cells were lysed 48 h after transfection, whereas virion particles released in the supernatant were first purified by ultracentrifugation through a 25% sucrose cushion. Both cellular and viral lysates were analyzed by WB. **(D)** HEK293T cells were ectopically transfected with DNAs coding for the indicated IFITM3s along with the G protein of the Vesicular Stomatitis virus fused to the mEmerald fluorescent reporter (mEm.-VSV-G), before confocal microscopy analysis 24 h later. Representative confocal microscopy images and graph presenting the proportion of double-positive cells displaying perinuclear accumulation of mEm.-VSV-G (binary scoring; three independent experiments; between 50 and 100 cells scored per sample). *P-value < 0.0001 following a one-way ANOVA, Tukey's multiple comparisons test; non-statistically significant differences, not shown. **(E)** Lysates obtained from cells expressing VSV-G along with control, WT and mutant IFITM3 were either untreated or treated with EndoH or PNGaseF before WB and densitometry quantification of the EndoH-sensitive and -resistant VSV-G forms. The scheme presents a simplified view of the trafficking and the modifications of glycoproteins during their transit through the ER–Golgi system. As EndoH is able to digest simple oligosaccharides added by enzymes resident in the ER and *cis*-Golgi, but not complex ones added later in the *medial/trans* Golgi compartment, differential susceptibility to EndoH provides a robust biochemical method to assess glycoproteins progression through the ER–Golgi. Panels present representative results, whereas the graph presents the proportions of EndoH-sensitive and -resistant VSV-G obtained in five independent experiments. *P-value of 0.0021 following a one-way ANOVA, Tukey's multiple comparisons test over control. Uncropped blots and source data are provided in the relevant section.
Source data are available for this figure.

domain regulates also Golgi trafficking of IFITM1, as well as of the more distant PRRT2 protein for which mutations in this exact domain have been identified in patients affected by PKD. By comparing the behavior of mutant and WT IFITM3, our results indicate that correct exit from the Golgi is a novel and important step in the biology of IFITM3 because mutations or high levels of expression of the wild-type lead to the protein's retention in this organelle and drive general dysfunctions of the glycoprotein secretory pathway. These defects can be rescued by addition of Amphotericin B, an antifungal compound known to relieve membrane fusogenicity

defects driven by IFITMs, as well as by overexpression of the v-SNARE GS15, overall supporting a model in which when IFITMs are retained in the Golgi, they affect the secretory pathway likely by interfering with v- to t-SNARE vesicles fusion. Differences in VSV-G trafficking in WT versus IFITMs-KO cells support a previously unappreciated physiological role for IFITMs in the regulation of the ER–Golgi secretory pathway, which can be especially important in conditions that favor high IFITMs expression, as for example, IFN stimulation. Overall, our study identifies a novel domain that regulates the egress from the Golgi of multiple members of the

dispanin/CD255 family and in the case of IFITM3 it highlights the importance that this step bears for the regulation of the cellular glycoprotein secretory pathway.

# Results

### An IFITM3 mutant affects glycoprotein trafficking

Using an extensive series of IFITM3 mutants generated by the Brass' laboratory (13), we previously identified a mutant in the CIL region of IFITM3 (CIL, mutant 85-90 in which six consecutive residues are mutated to alanine, Fig 1A) that exerted a strong inhibitory effect on the incorporation of HIV-1 Envelope glycoproteins on released virion particles (14). At its simplest, the formation of infectious HIV-1 virion particles requires the gathering on the plasma membrane of two classes of structural viral proteins: Gag (and Gag-Pro-Pol, not marked here for clarity's sake) and Env (Fig 1B). Whereas Gag is translated from cytoplasmic mRNAs from free ribosomes, the Envelope glycoprotein undergoes co-translational ER translocation and reaches the plasma membrane following the ER–Golgi secretory pathway (15). As such, loss of Envelope incorporation in virion particles may reflect potential effects of IFITM3 along this axis that we decided to investigate further.

Indeed, when expressed in HEK293T cells undergoing HIV-1 virion assembly (Fig 1B and C), the 85-90 IFITM3 mutant induced only a moderate, yet detectable, decrease in the levels of cell-associated Gag and Env proteins when compared with control or WT IFITM3–expressing cells. In contrast, purified virions exhibited a slight decrease in the levels of mature p24 capsid protein and a drastic reduction in the levels of Env (Fig 1C). At the cellular level, this was accompanied by an accrued perinuclear accumulation of HIV-1 Env in the presence of the 85-90 IFITM3 mutant (Fig S1). From these starting observations, we decided to determine whether this defect could be more generally reflective of an interference with Golgi-mediated trafficking, independently from HIV-1. To this end, the 85-90 IFITM3 mutant was co-expressed along with a fluorescent G protein of the Vesicular Stomatitis Virus (mEm.-VSV-G, Fig 1D), before confocal microscopy analysis. Under these conditions, a strong perinuclear accumulation of mEm.-VSV-G was observed in cells expressing the mutant IFITM3, overall suggesting an involvement of the Golgi apparatus (Fig 1D).

To further support this result, we analyzed the EndoH susceptibility of a non-tagged VSV-G protein followed by WB analysis, as an independent and commonly used technique to appreciate the progression of glycosylated proteins from the ER through the Golgi (Fig 1E). Glycans are added in an orderly fashion to proteins that engage in the secretory pathway as they move through Golgi cisternae. Whereas simpler glycans that are susceptible to EndoH digestion are added in the *cis*-Golgi, more complex ones that cannot be processed by EndoH are added later in the *mid/trans*-Golgi compartment (as depicted schematically in Fig 1D, and contrarily to PNGase F that instead cleaves all glycans). As such, changes in the proportion of EndoH-sensitive/resistant protein forms reflect *cis*-Golgi trafficking defects.

Under these conditions, expression of WT IFITM3 led to a small but detectable accumulation of EndoH-sensitive VSV-G when

compared with control cells which however did not reach statistical significance, under the conditions used here. Instead, the proportion of EndoH-sensitive VSV-G protein was significantly increased upon expression of the 85-90 IFITM3 mutant.

Thus, both the biochemical and the confocal microscopy analyses of the accumulation of the VSVg glycoprotein, indicate that the 85-90 IFITM3 mutant induces a strong defect in the ER–Golgi secretory pathway.

To determine whether this effect was specific for viral glycoproteins, or could also affect cellular ones, we examined the behavior of CD93 (16), a cell surface glycoprotein involved in cell adhesion and apoptotic cells clearance. As expected, CD93 was essentially concentrated at the plasma membrane when expressed alone (Fig S2). However, co-expression with the 85-90 IFITM3 mutant was sufficient to induce perinuclear relocalization of a detectable fraction of CD93, similarly to what observed for VSV-G.

Overall, these results indicate that the expression of the 85-90 IFITM3 mutant affects the normal functionalities of the ER–Golgi secretory pathway and general glycoprotein trafficking.

### The 85-90 IFITM3 mutant accumulates in and leads to structural changes of the Golgi apparatus

The intracellular distribution of the 85-90 IFITM3 mutant did not significantly differ from WT with respect to several markers (Fig S3). However, it exhibited an accrued accumulation at the *cis*-Golgi over *wild-type* IFITM3 (GM130 marker, Fig 2A and B for a 3D-reconstruction of positive cells). Most interestingly, the 85-90 IFITM3 mutant induced also gross morphological changes in the Golgi apparatus that became largely inflated, as opposed to its punctiform appearance in control and WT IFITM3–expressing cells (Fig 2A, colocalization in yellow). These structural changes were not only apparent after 3D-reconstruction of positive cells following confocal microscopy analysis (Fig 2B, colocalization in grey), but also after electron microscopy analysis which more precisely indicated that the shape of the Golgi apparatus had lost its classical punctiform distribution in favor of an expansion of large vesicles (Fig 2C).

### The 85-90 mutation identifies a novel domain within IFITM3 that regulates the protein's egress from the Golgi

To start determining whether the 85-90 mutation altered normal and yet unknown aspects of the biology of IFITM3, or endowed it with novel properties, we first assessed whether the 85-90 mutation induced the retention or the relocalization of IFITM3 at the Golgi. To this end, we performed a time-course analysis of both WT and mutant IFITM3 from early to late time points after ectopic DNA transfection and protein translation (Fig 3). Although both proteins exhibited a predominant Golgi distribution at 6 h post transfection, such concentration gradually diminished in the case of WT IFITM3, indicating that under normal conditions, IFITM3 transits from the Golgi and then exits it. On the contrary, no significant changes were observed in the case of the 85-90 IFITM3 mutant that remained colocalized in the Golgi over time, indicating that the 85-90 mutation affects the natural process of egress of IFITM3 from the Golgi, rather than inducing a de novo behavior.

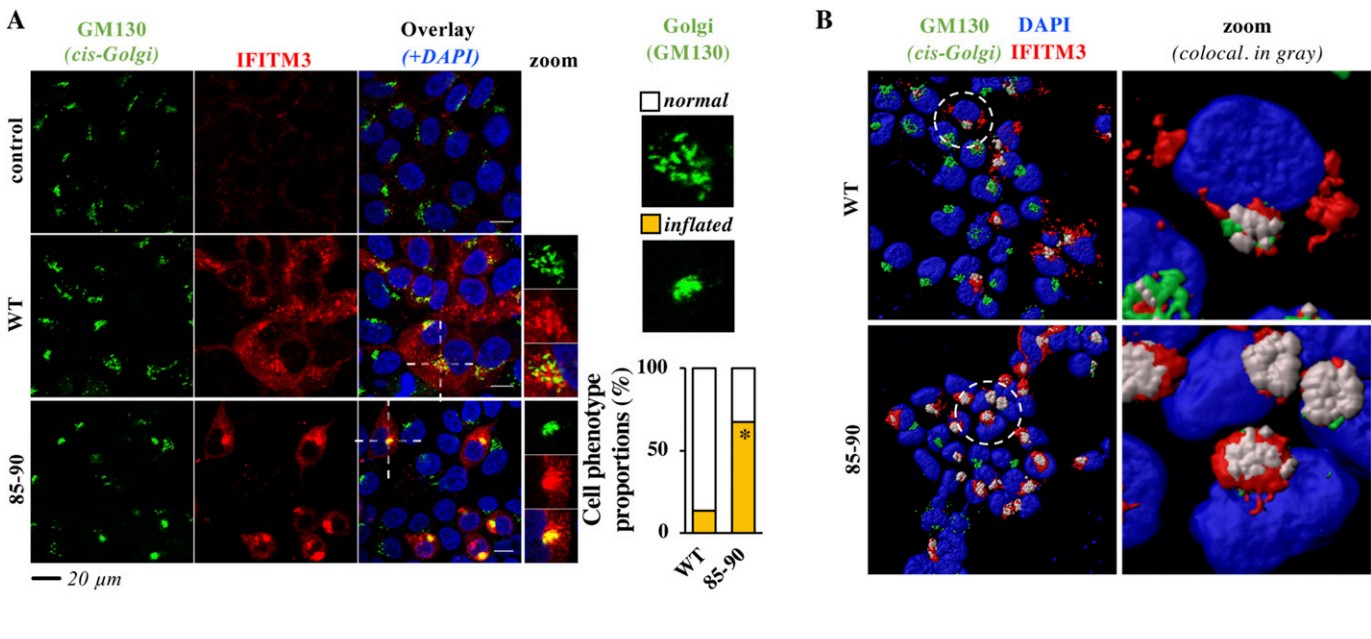

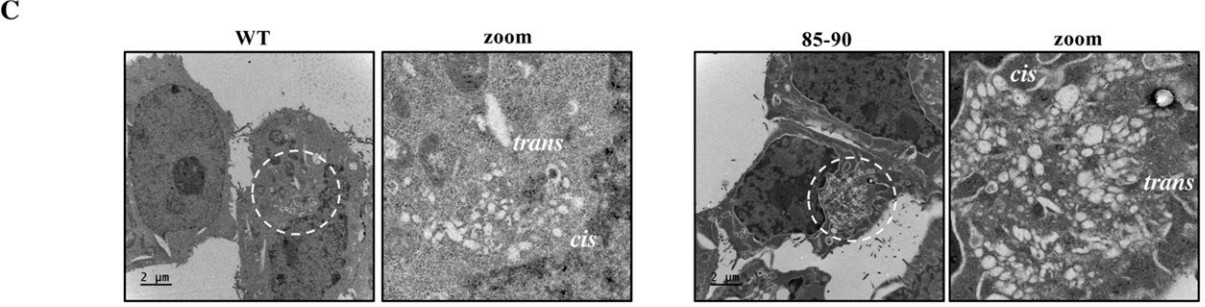

**Figure 2. The 85-90 IFITM3 mutant colocalizes with and induces gross morphological changes in the Golgi.**
**(A, B)** Confocal microscopy analysis and 3D-reconstruction of cells expressing the indicated IFITM3 proteins. Colocalization of GM130 and IFITM3 is shown in yellow and grey, respectively. Representative images and graph presenting phenotype proportions in at least 100 cells scored per condition (binary scoring: inflated or normal Golgi, three independent experiments). *P = 8.3 × $10^{-9}$ following an unpaired, two-tailed *t* test. **(C)** Representative electron microscopy analyses of HEK293T cells expressing WT or 85-90 IFITM3 proteins. The region enlarged at the right of each panel corresponds to the red inset; *cis* and *trans* regions of Golgi marked. Source data are provided in the relevant section. Source data are available for this figure.

In further support of this notion, incubation of cells with the broad trafficking inhibitor Monensin induced accrued Golgi accumulation of WT IFITM3, but did not modify the distribution pattern of the 85-90 mutant (Fig S4), in agreement with the contention that the 85-90 mutation negatively affects a protein domain within the CIL that regulates the normal trafficking of IFITM3 through the *cis*-Golgi.

### WT IFITM3 also accumulates in the Golgi when expressed at high levels ectopically, or upon prolonged IFN stimulation and it modulates glycoprotein trafficking

Expression of WT IFITM3 to levels comparable with the 85-90 mutant did not lead to the same drastic defects in Golgi trafficking. However, we do not believe this is surprising in light of the fact that the 85-90 IFITM3 is completely impaired in Golgi exit. To more precisely determine the extent to which the defects of this mutant related to *wild-type*, we first ectopically expressed increasing levels of WT IFITM3 along with mEm.-VSV-G in HEK293T cells (Fig 4A). Under

these conditions, increased expression of WT IFITM3 led to a proportional increase in the perinuclear accumulation of both proteins, similarly to what observed for the 85-90 IFITM3 mutant, indicating that this mutation exacerbates a behavior that can be observed upon high levels of expression of WT IFITM3. To determine whether this could also be observed with endogenous proteins, we determined the intracellular distribution of endogenous IFITM3 upon IFNα stimulation of human primary blood lymphocytes (PBLs), monocyte-derived macrophages and A549 cells, a lung-derived cell line in which IFITM expression is also IFN-sensitive. Furthermore, we analyzed HeLa cells that express constitutively high levels of IFITMs and their corresponding IFITMs-KO derivative in which IFITM1, 2 and 3 have been ablated by CRISPR/Cas9 (17). In all cell types examined and independently from the basal levels of IFITM3 expression (e.g., IFITM3 is clearly detectable in stimulated PBLs even in the absence of IFN), IFN stimulation led to a significant increase in the accumulation of endogenous IFITM3 in the *cis*-Golgi (Figs 4B and S5 for Pearson's coefficients).

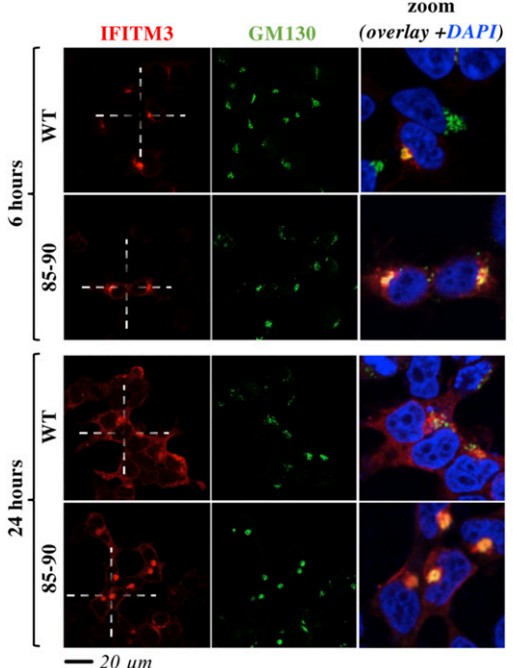

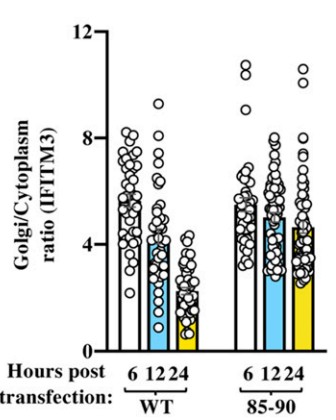

| | | 6 | 12 | 24 |
|---|---|---|---|---|
| WT 6hrs vs | WT | | * | **** |
| | 85-90 | ns | ns | * |
| 85-90 6hrs vs | WT | ns | ns | **** |
| | 85-90 | | ns | ns |

*one-way Anova, Tukey's multiple comparison test*

**Figure 3. Time-course confocal microscopy analysis indicates that the 85-90 IFITM3 mutation affects the normal egress of IFITM3 from the Golgi.**
HEK293T cells were examined by confocal microscopy at different times post ectopic DNA transfection to determine the localization of IFITM3 proteins over time. Representative pictures and graph presenting the distribution of IFITM3 in the Golgi (calculated as a Golgi/cytoplasm ratio on 39 to 62 cells per time point and per condition in two independent experiments, AVG, SEM and individual values). The table presents $P$-values obtained after a one-way ANOVA, Tukey's multiple comparisons test between the indicated conditions: ns, nonsignificant; *$P$-value < 0.05; ****$P$-value < 0.0001. Source data are provided in the relevant section.
Source data are available for this figure.

To determine whether accumulation of WT IFITMs at the Golgi could exert a modulatory role in glycoproteins trafficking, we used HeLa WT and IFITMs-KO cells as these cells allowed us to examine the effects of endogenous IFITMs in VSV-G glycoprotein trafficking, in the absence of pleiotropic effects due to IFN stimulation. Under these conditions, the VSV-G protein appeared qualitatively more distributed at the plasma membrane in IFITMs-KO cells than in WT HeLa cells by confocal microscopy (Fig 5A). Under the hypothesis that removal of IFITMs should result in higher levels of glycoproteins at the plasma membrane, living cells were labeled with an anti-VSV-G antibody to specifically recognize plasma membrane VSV-G, before fixation and quantification by flow cytometry analysis. Under these conditions, a significant increase in the levels of plasma membrane VSV-G was measured in IFITMs-KO cells when compared with WT cells (Fig 5B for representative panels, histograms and cumulative analysis of the median fluorescence intensity, MFI, of cell surface VSV-G-positive cells). Instead, no significant differences were observed in MFI after transfection of DNA coding a cytoplasmic GFP reporter (Fig 5B).

Overall and despite the caveat that we used cells in which all three IFITMs had been removed to avoid possible redundancies, these results indicate that endogenous IFITM3 do accumulate in the Golgi in conditions that promote its high levels of expression and, under these conditions, contribute to regulate glycoprotein trafficking, a novel aspect in the biology of these factors.

### Accumulation of the 85-90 IFITM3 mutant in the Golgi and IFITM3-driven glycoproteins trafficking defects can be relieved by Amphotericin B and v-SNARE overexpression

One of the key features of IFITM family members is their ability to interfere with viral-to cellular membranes fusion (18, 19, 20). We

thus hypothesized that accumulation of IFITM3 at this location could cause Golgi dysfunctions because of its ability to interfere with the ability of v-SNARE vesicles to fuse to t-SNAREs compartments, a process that is key to maintain protein and membrane fluxes and is key for the integrity of the Golgi. Our first approach was to use fluorescence resonance energy transfer between donor and acceptor of v- and t-SNAREs couples in living cells. However, the high toxicity of a SNARE fusion inhibitor control N-ethylmaleimide (*NEM*) in our system, prevented us from drawing firm conclusions with this technique. To circumvent this issue, we first used Amphotericin B (AmphoB), an antifungal compound known to relieve the membrane fusogenicity defects driven by IFITM3 (21). Incubation with AmphoB was sufficient to induce a strong relocalization of VSVg from the *cis*-Golgi to the plasma membrane in the presence of the 85-90 IFITM3 mutant and it also efficiently redistributed the mutant away from this compartment (Fig 6A), strongly suggesting that the 85-90 IFITM3 mutant is a functional IFITM3 molecule that drives a membrane fusion defect in the Golgi. Interestingly, similar changes were observed in the presence of high, but not low, levels of WT IFITM3, in agreement with the accrued perinuclear accumulation of WT IFITM3 observed before (Figs 6A and S6A for representative pictures of low concentrations of WT IFITM3). In further support of a previously unrecognized regulatory role of WT IFITM3 on Golgi trafficking, treatment of VSVg-transfected HeLa cells with AmphoB led to a significant increase in the cell surface expression levels of VSVg in WT cells that express high levels of IFITMs, but not in IFITMs-KO cells (Fig S6B).

v-SNAREs overexpression is also known to bypass membrane fusion defects between Golgi membranes (22, 23, 24) and we thus hypothesized it could relieve the defect imposed by the 85-90 IFITM3 mutant. To this end, increasing levels of GS15 were expressed along with a constant amount of both 85-90 IFITM3 mutant and

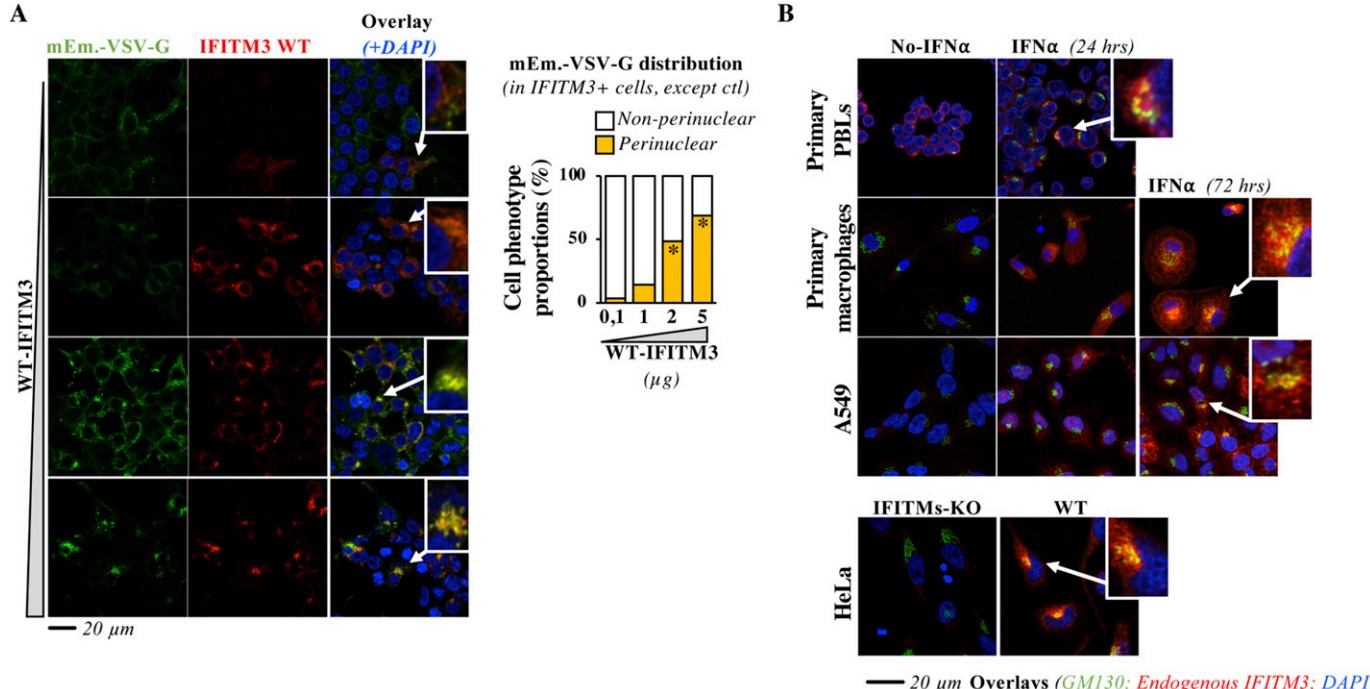

**Figure 4. WT IFITM3 also accumulates in the cis-Golgi upon high levels of expression or prolonged IFN stimulation.**
**(A)** HEK293T cells were transfected with a constant amount of DNA coding mEm.-VSV-G and increasing levels of DNA coding WT IFITM3, prior confocal microscopy 24 h later. Representative pictures and graph presenting the proportion of IFITM3-positive cells exhibiting normal or perinuclear accumulation of mEm.-VSV-G (binary scoring; two independent experiments; between 44 and 167 cells scored per sample). A magnified view of the indicated cells is provided in the inset. *P-value < 0.0001 following a one-way ANOVA, Dunnett's multiple comparisons test against the lowest dose of WT IFITM3 (0,1). **(B)** Purified primary blood lymphocytes were stimulated for 24 h with anti-CD3 and anti-CD28 antibodies plus 150 U/ml of IL2, whereas primary monocytes were differentiated into macrophages upon incubation with M-CSF for 4 d. Both primary blood lymphocytes and macrophages were then stimulated for the indicated times with 1,000 U/ml of IFNα2 to stimulate IFITMs expression, before confocal microscopy analysis. Representative pictures from experiments carried out with cells of three different donors. A549 cells were similarly treated, whereas HeLa cells that express IFITMs constitutively were directly analyzed. Pearson's coefficients were determined for primary macrophages and A549 cells and are provided in extended data section, Fig 4.
Source data are available for this figure.

VSV-G, to determine whether this could lead to Golgi trafficking normalization (Fig 6B). Under these conditions, GS15 was able to relieve the perinuclear accumulation defect of the VSV-G glycoprotein induced by the 85-90 IFITM3 mutant and was also able to act similarly on the IFITM3 mutant itself, supporting, although not directly proving, our hypothesis that a main disruptive event due to the accumulation of the 85-90 IFITM3 mutant in the Golgi is the inhibition of v- to t-SNAREs fusion.

### Identification of the domain mediating the egress of WT IFITM3 from the Golgi

To more precisely map the domain involved in the exit of IFITM3 from the Golgi, individual amino acids spanning the entire CIL were mutated to alanine and mutants were analyzed by confocal microscopy (Figs 7A–C and S7, alanine residues in the CIL were instead mutated to glycines). With the exception of two mutants that were barely detectable by WB (D92A and G95A), the remaining mutants were expressed to detectable levels upon WB analysis. We noticed that the D92A mutant acquired also a nuclear localization unusual for IFITM proteins, but this was not characterized further. This mutagenesis study identified a number of residues the mutation of which led to accrued accumulation of IFITM3 into the Golgi (S81, V82,

K83, S84, R85, D86, K88, A96, Q97, A98, A103, K104, C105, and N107), indicating that several residues within the CIL contribute to this phenotype. Interestingly, however, the mutations that resulted in levels of IFITM3 localization in the Golgi equivalent to those of the 85-90 mutant were essentially clustered in a patch (from residues S81 to D86) posited at the boundary between the end of the intramembrane domain of IFITM3 (IMD) and the start of the CIL and overlapping with the 85-90 stretch.

Thus, the complete mutagenesis of the CIL allowed us to more precisely refine a core domain responsible for the egress of IFITM3 from the Golgi ($_{81}$SVKSRD$_{86}$).

### A functional Golgi egress domain is conserved among members of the dispanin/CD225 subfamily A, as well as in PRRT2, a member of the dispanin/CD225 subfamily B, genetically associated to PKD

IFITMs belong to the larger dispanin/CD225 family which is itself divided into four subfamilies (A to D, Fig 8A) (2). The A subfamily is composed of the IFN-inducible and antiviral IFITM1, 2 and 3, as well as of IFITM5 and IFITM10, which are not IFN regulated and that in the case of IFITM5 are associated to Osteogenesis imperfecta type V, a bone-related genetic disease (4). We noticed that the SVKSRD domain was perfectly conserved among human antiviral IFITMs and

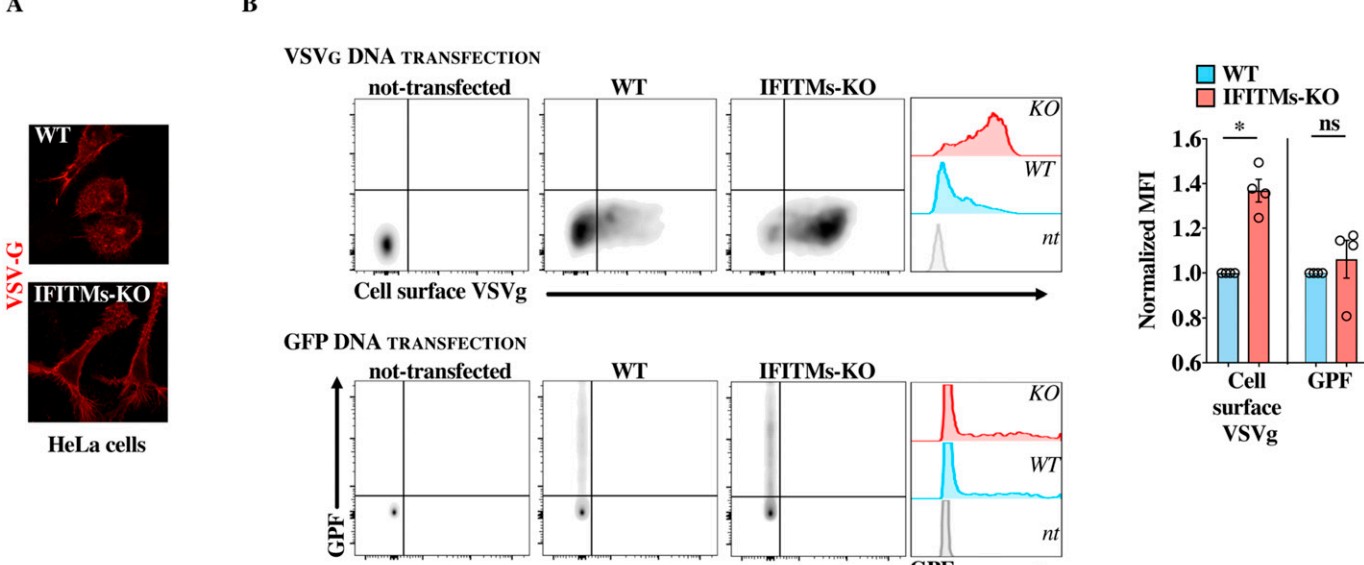

**Figure 5. WT IFITM3 accumulation in the cis-Golgi alters glycoproteins trafficking.**
**(A)** WT and IFITM1, 2, 3 KO cells were transfected with DNA coding for VSVg before confocal microscopy analysis (qualitative analysis and representative pictures obtained). **(B)** HeLa WT and IFITMs-KO cells were either transfected with DNA coding for a GFP marker or for VSVg before flow cytometry analysis. To quantify the amount of VSVg arriving at the plasma membrane, cells were labeled with an anti-VSVg antibody, before fixation and analysis. Typical panels obtained and overall changes in the median fluorescent intensity (MFI) in the different conditions. * and ns, statistically significant and nonsignificant *P*-values after unpaired two-tailed *t* tests between the indicated conditions. Source data are provided in the relevant section.
Source data are available for this figure.

varied to a different extent among members of the B, C and D subfamilies (Fig 8A, conserved residues in bold).

To determine whether this domain also controlled Golgi egress in other IFITM proteins belonging to the A subfamily, a single point mutation was introduced in the corresponding domain of IFITM1 (R64A), which contrarily to IFITM3 and to IFITM2 is essentially localized at the plasma membrane. Under these conditions, the introduction of a single point mutation was sufficient to drive the redistribution of IFITM1 from the plasma membrane to the Golgi (Fig 8B), indicating that this domain is also controlling Golgi trafficking of IFITM1.

Despite the fact that the functions of certain dispanin/CD225 family members remain unknown (as is the case for TMEME90A, TMEM91, TMEM233, and TMEM265), the information existing on others (TUSC5, TMEM90B, and PRRT2) indicate functions revolving around vesicular trafficking and vesicular behavior (5, 25, 26). Among them, PRRT2 acts as a regulator of neurotransmitter vesicles in neuron cells and is genetically linked to PKD (the most common type of paroxysmal movement disorder), as well as benign familial infantile seizures. PRRT2 shares three conserved residues with the corresponding domain of IFITM3 and we noticed that two mutations in PRRT2 have been identified in patients affected by PKD in the same region: A291V and R295Q, corresponding to the underlined positions 1 and 5 of the SVKSRD domain of IFITM3 (27, 28). So, we decided to analyze the intracellular distribution of these mutant proteins along with an additional alanine mutant (R295A) (Fig 8C). Whereas WT PRRT2 exhibited a predominant plasma membrane distribution, introduction of the abovementioned mutations led to accrued redistribution of PRRT2 in the Golgi, similarly to what

described for IFITM3. Thus, these results indicate that this region is also important for the regulation of the trafficking of PRRT2 from the Golgi.

# Discussion

In the present study, through the careful characterization of the 85-90 mutant, we have defined a novel domain that regulates the normal egress of IFITM3 from the Golgi and that similarly regulates the trafficking of IFITM1 and PRRT2, the other members of the dispanin/CD255 family tested. Whereas ER-sorting protein domains have been identified (29), similar ones have not been characterized for the Golgi, and our results thus provide an important piece of the puzzle to unravel how proteins that transit through the Golgi may be sorted out of it.

The Golgi egress domain of IFITM3 can either act as a docking site for specific trafficking co-factors (for instance Rab proteins), it can indirectly regulate the functions of other IFITM regions themselves involved in trafficking, or of course a combination of both.

The $_{81}$SVKSRD$_{86}$ domain is situated between the IMD and the beginning of the CIL and it is thus in close proximity with several features of importance for the functions of IFITM3: a short amphipathic helix (30); two cysteine residues that can be palmitoylated and thus affect IFITM association to membranes (C71, C72 in human IFITM3; a third cysteine C105 is also present that appears less important in IFITM3 functions (31, 32)) and a G-x$_3$-G oligomerization domain (33).

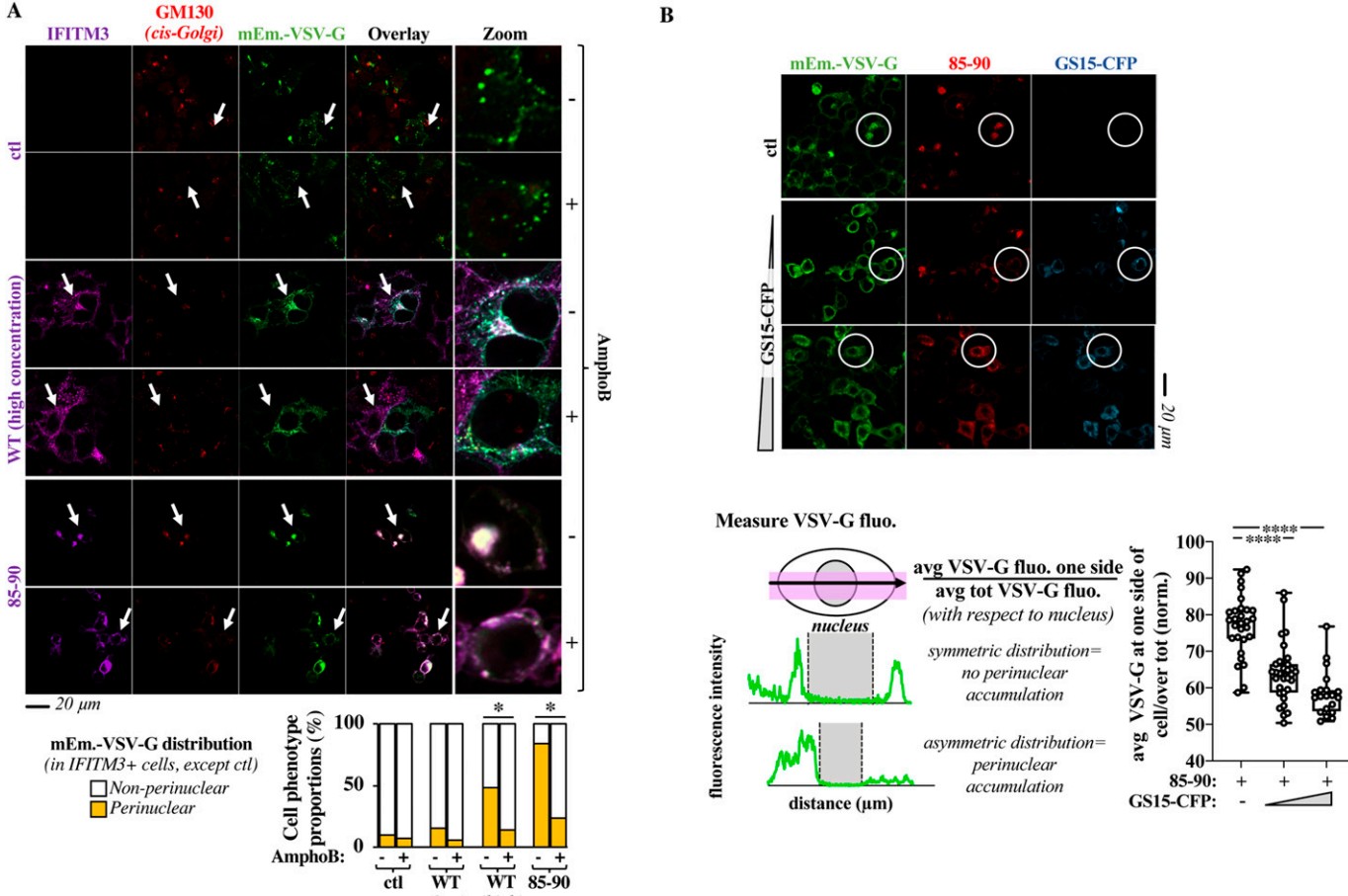

**Figure 6. The Golgi trafficking defect observed in the presence of the 85-90 IFITM3 mutant or of high levels of WT IFITM3 is relieved by Amphotericin B and v-SNARE overexpression.**
**(A)** HEK293T cells transfected with mEm.-VSV-G and 85-90 IFITM3 (or of WT IFITM3 expressed at low or high levels: 1 and 4 μg, representative pictures of WT IFITM3 at low concentrations are provided in the Fig S6A) coding DNAs were either mock treated or treated with AmphoB at 1 μM for 18 h before confocal microscopy analysis. Representative images along with the cumulative analysis of the proportion of cells presenting the indicated VSV-G distributions (from 40 to 100 cells scored per condition in three independent experiments). *P-values of 0.002 and <0.0001 for WT-high and 85-90, respectively, after unpaired two-tailed t tests between the indicated conditions. **(B)** HEK293T cells expressing constant amounts of 85-90 IFITM3 and mEm.-VSV-G were co-transfected with increasing amounts of GS15-CFP coding DNA, before confocal microscopy analysis (GS15/85-90 ratios of 0.2 and 0.5). Given that VSV-G accumulation in the Golgi leads to its asymmetrical perinuclear distribution in the cell, VSV-G fluorescence was determined for each cell according to the presented scheme. The fluorescence measured over distance in the cell was used to calculate the proportion of VSV-G protein present at one side (the left side was set as the side with higher protein accumulation). Representative pictures and graph presenting the distribution of VSV-G at one side of the cell in triple-positive cells (n = 2 with 20–31 cells analyzed per condition). White circles highlight examples of phenotypic differences in the perinuclear accumulation of VSV-G. ****P-value of <0.0001 following a one-way ANOVA, Tukey's multiple comparisons test over 85-90 IFITM3 condition with no GS15-CFP. Source data are provided in the relevant section.
Source data are available for this figure.

Given that the deleterious effects of the 85-90 IFITM3 mutant can be counteracted by Amphotericin B, we believe this mutant is a proficient membrane fusion inhibitor and thus unlikely to affect the functionality of the amphipathic helix and G-$x_3$-G domains. Instead, mutations in this domain could alter association with Rabs, or modulate access of proximal domains to post-translational modifications that in turn affects IFITM trafficking. Among them, an effect on the extent of palmitoylation at proximal cysteines would be in agreement with a recent model proposed by the Rothman's lab for proteins exit from the Golgi based on a palmitoylation gradient (34). This hypothesis is of interest because two cysteines proximal to the Golgi exit domain identified here are present in numerous members of the dispanin/CD255 family:

C276/C278 in PRRT2; C114/116 in TUSC5 as members of the B subfamily; C171/C172 in TMEM90A; C015/C106 in TMEM91 for members of the C subfamily and C152/C153 for PRRT1, member of the D subfamily. Whether this is the case remains for the moment to be experimentally proven. To add to the complexity of this region, the K83 residue can be ubiquitinated (6) and together with residue K104 it has been recently described to participate in PIP3 binding of IFITM3 at the plasma membrane of B cells (11). Whereas our confocal microscopy data suggest that the K83 IFITM3 mutant may not act as a functional PI3K platform simply because it is retained in the Golgi, ubiquitination of the $_{81}$SVKSRD$_{86}$ domain could be important to finely tune the transit of these proteins through this organelle.

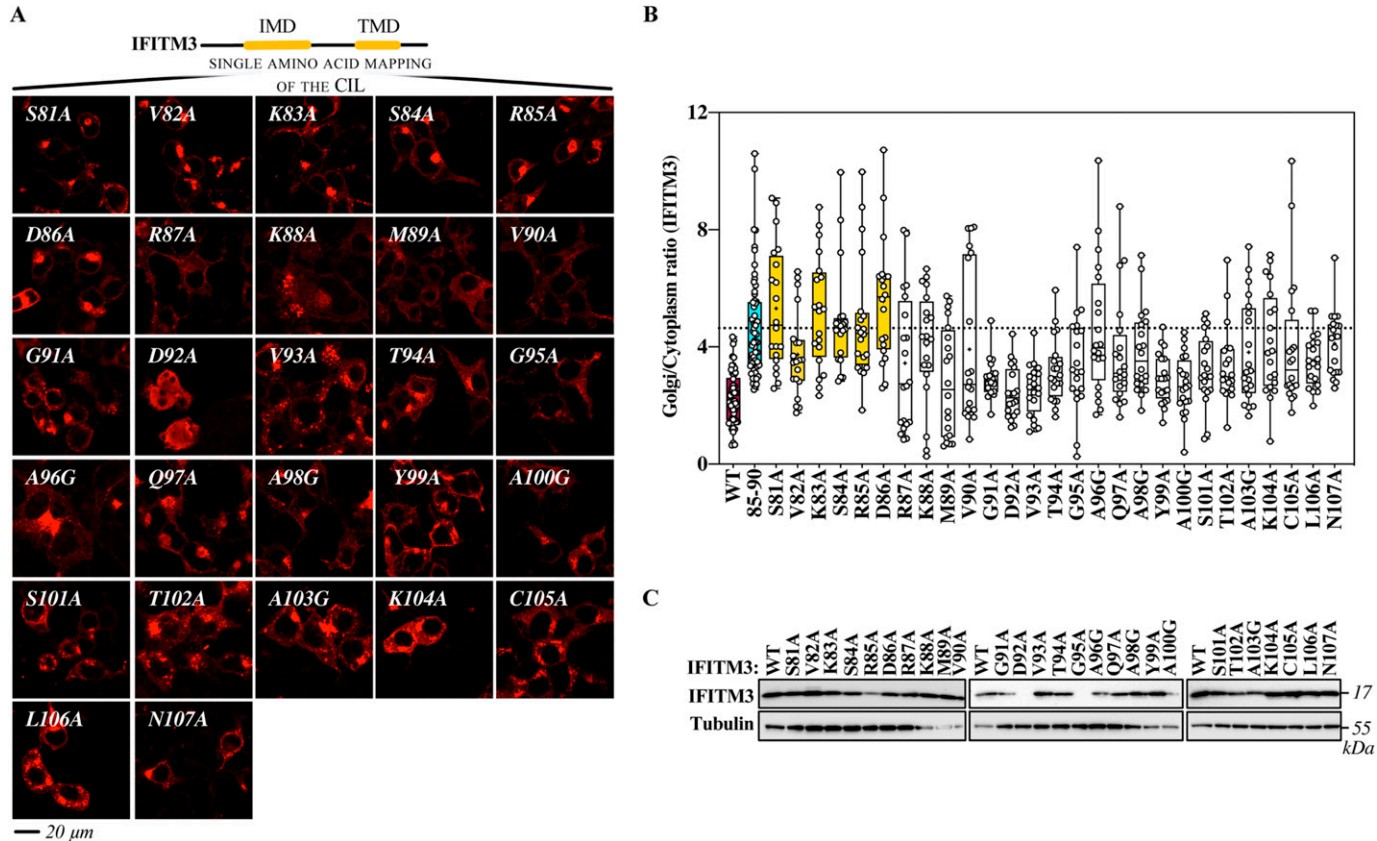

**Figure 7. Mutagenesis of the entire cytoplasmic intracellular loop of IFITM3 finely maps the domain involved in the egress of IFITM3 from the Golgi.**
**(A)** Individual amino acid of the cytoplasmic intracellular loop were changed to alanine or to glycine when alanine residues were present, before confocal microscopy analysis. Representative pictures are shown here (IFITM3 only), whereas their colocalization with the cis-Golgi marker GM130 is shown in the extended data Fig 3.
**(B)** Quantification of the IFITM3 proportion in Golgi is provided as a Golgi/cytoplasm ratio in the box and whisker plot (20 cells per mutant in two to three independent experiments analyzed). Asterisks and lines within the box indicate averages and median values, respectively. Yellow boxes indicate mutants with Golgi/cytoplasm IFITM3 ratios equivalent to the 85-90 IFITM3 mutant and non-statistically significant when compared with 85-90 following an ordinary one-way ANOVA, Dunnett's multiple comparisons test. **(C)** Representative WB analysis of mutant proteins. Uncropped blots and source data are provided in the relevant section.
Source data are available for this figure.

## Which are the consequences of an increased retention of these proteins at the Golgi?

PRRT2 is a neuronal specific protein involved in pre- and post-synaptic neurotransmitter vesicles regulation and has been genetically linked to PKD. Whereas most PKD patients suffer from mutations that result either in absent or severely truncated proteins, a few exhibit non-synonymous changes in the context of the full-length protein. Our results indicate that among them, mutations in the Golgi egress domain may lead to the retention of PRRT2 in the Golgi, thus diminishing the concentration of active protein at its normal functional site, which is the plasma membrane.

In the case of IFITM3, complete retention of the 85-90 mutant bears dramatic consequences for the functionalities of the Golgi and for the glycoprotein secretory pathway, likely because of the fact that the protein is massively concentrated at this location and interferes, as IFITMs do, with membrane fusion events. A significant proportion of WT IFITM3 also accumulates in the *cis*-Golgi upon ectopic expression or IFN stimulation in all cells tested. In this case,

given the presence of a functional Golgi egress domain, we hypothesize that this stalling may be due to the saturation of the Golgi exit process in conditions that favor high levels of expression of IFITM3. Nonetheless, despite lower levels of accumulation at this location, the differences observed in VSV-G trafficking between WT and IFITMs-KO HeLa cells clearly indicate that IFITM molecules do play a modulatory role in the secretory pathway and in this respect could explain past observations on decreased levels of expression of viral glycoproteins (35, 36). The importance and the consequences of IFITM3-mediated modulation of Golgi functions remain to be determined.

IFITMs can represent novel physiological modulators of the ER–Golgi secretory pathway and contribute to complex cellular responses, in addition to their role as direct viral inhibitors. Alternatively, these could represent deleterious side effects consequent to the expression of a general membrane fusion inhibitor in the cell. The literature provides other examples of the ambivalence of antiviral restriction factors, as for example, the family of apolipoprotein B mRNA editing enzyme, catalytic polypeptide-like 3 proteins (APOBEC3), prototypical anti-retroviral defense proteins.

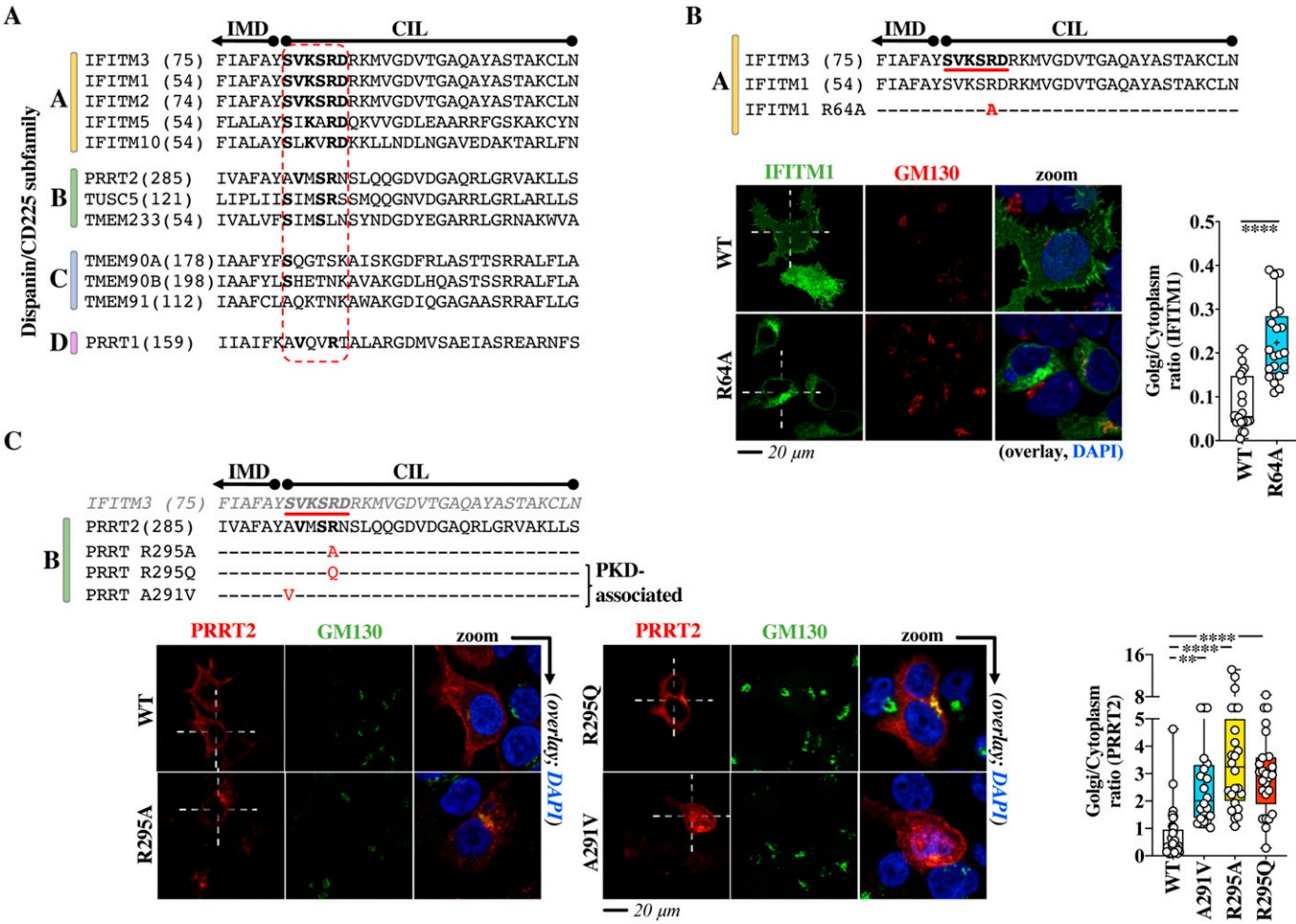

**Figure 8. The Golgi egress domain of IFITM3 is conserved across vertebrate members of the dispanin/CD225 subfamily A and is functionally conserved in PRRT2, a member of the B subfamily as shown by genetic mutations associated with paroxysmal kinesigenic dyskinesia.**
**(A)** Alignment of the indicated portions of human members of the different dispanin/CD225 subfamilies. The relevant domain is circled in red. Bold indicates conserved residues. The position of the first amino acid of each sequence is shown within parentheses. **(B)** Single point mutation and confocal microscopy analysis of the Golgi egress domain of IFITM1. **(C)** Point mutations and confocal microscopy analysis of PRRT2 mutants in the region corresponding to the Golgi egress domain of IFITM3. R295Q and A291V are genetic mutations associated to paroxysmal kinesigenic dyskinesia. Representative pictures are shown for each protein. Whiskers and boxes plots of the quantification of the proportion of each protein in the Golgi as a Golgi/cytoplasm ratio (25–75 percentiles with individual cells represented as dots; group average indicated by an asterisk, n = 20 in three independent experiments). Colored boxes represent statistical significant differences after a two-tailed $t$ test or a one-way ANOVA, Dunnett's multiple comparisons test (B and C, respectively) of the examined mutant over WT. \*\*$P$-value of 0.0019; \*\*\*\*$P$-value < 0.0001. Source data are provided in the relevant section.

Whereas the main function of APOBEC3 members is to mutate and inactivate retroviral genomes thanks to their cytidine deaminase activity, this same enzymatic activity also contributes to the tumorigenic process when it is turned against the cellular genome (37, 38, 39).

In conclusions, several reports indicate that IFITMs can play pleiotropic functions: they interfere with trophoblast fusion during placental formation by affecting syncytin-mediated membrane fusion (12, 40); they lead to glucose-related metabolic dysfunctions in mice through unknown mechanisms (41) and they concur to B cell signaling (11). Our study reveals that IFITMs also intersect the functionalities of the secretory pathway and may thus play ambivalent roles in either global cellular responses to infection or else contribute to drive Golgi-related dysfunctions in certain pathological conditions in which their expression appears deregulated, as for example, in cancer or in interferonopathies.

# Materials and Methods

### Cells, plasmids and antibodies

Human embryonic kidney cells (HEK293T, ATCC Cat. no. CRL-3216) were maintained in complete DMEM media with 10% Fetal Calf Serum (Cat. no. F7524; Sigma-Aldrich). Control and IFITM1, 2, 3 knockout A549 and HeLa cells obtained by CRISPR/Cas9 gene editing were a kind gift from Howard Hang, Rockfeller University (17). Untagged WT IFITM3 (gene ID: 10410) and IFITM1 (gene ID: 8519) and corresponding mutants were either previously described (85-90

(13,14)), or cloned (all others) by standard mutagenesis techniques in the plasmid pQCXIP (Clontech). WT human PRRT2 (gene ID: 112476) was synthesized as N-terminal HA tag fusion protein (Genewiz) in pcDNA3 vector (Thermo Fisher Scientific) and single point mutants were generated on this matrix by standard molecular biology techniques.

The HIV-1 proviral clone NL4-3 and the VSV-G expression constructs have been described elsewhere (42). The mEmerald-VSV-G (mEm.-VSV-G) coding construct was a gift from Michael Davidson (54307; Addgene); pcDNA-D1ER coding for an ER targeted enhanced CFP was a gift from Amy Palmer & Roger Tsien (cat. 36325; Addgene (43)). GS15-CFP (cyanin fluorescent protein) and HA-CD93 coding constructs were obtained from Dr. Oleg Varlamov from the Oregon National Primate Research Center (44) and Dr. Bryan Heit from the Western University (16), respectively. The following primary antibodies were used for WB or confocal microscopy, as indicated. Mouse monoclonals: anti-α-Tubulin, anti-HA, and anti-VSV-G (Cat. no. T5168, Cat. no. H3663 and Cat. no. V5507; Sigma-Aldrich, respectively), anti-LAMP2 (Cat. no. sc-18822; Santa Cruz Biotechnology), anti-CD63 and anti-GM130 (Cat. no. 556019 and Cat. no. 610823; BD Biosciences); anti-HIV-1 p24 (obtained through the NIH HIV Reagent Program, Division of AIDS, NIAID, NIH, contributed by Dr. Bruce Chesebro and Kathy Wehrly, Cat. no. ARP-3537). Rabbit polyclonal antibodies: anti-IFITM3 (Cat. no. 11714-1-AP; Proteintech), anti-IFITM1 (Cat. no. 60074; Proteintech), and anti-GM130 (Cat. no. Ab52649; Abcam). The sheep polyclonal antibody anti-HIV-1 Env was obtained through the NIH HIV Reagent Program, Division of AIDS, NIAID, NIH (contributed by Dr. Michael Phelan, Cat. no. ARP-288).

The following secondary antibodies were used for WB: anti-mouse, anti-rabbit, and anti-sheep IgG-Peroxidase conjugated (Cat. no. A9044 and Cat. no. AP188P; Sigma-Aldrich and Cat. no. P0163; Dako), whereas the following ones were used for confocal microscopy: donkey anti-rabbit IgG–Alexa Fluor 594 conjugate and donkey anti-mouse IgG–Alexa Fluor 488 conjugate (Cat. no. A-21207 and Cat. no. A-21202; Life Technologies).

Primary monocytes and lymphocytes were purified from the blood of healthy donors as described (45). White blood leukocytes were first purified through successive Ficoll and Percoll gradients. An enriched monocytes fraction was harvested at the Percoll interface and monocytes were purified by negative depletion (monocyte isolation kit II, Cat. no. 130-091-153; Miltenyi; purity equal/superior to 90%). Monocytes were differentiated into macrophages upon incubation for 4 d with human Macrophage-Colony Stimulating Factor (M-CSF at 100 ng/ml, Cat. no. 01-A0220-0050; Eurobio). PBLs were instead retrieved at the bottom of the Percoll gradient and stimulated for 24 h with 1 µg/ml each of anti-CD3 and anti-CD28 antibodies (Cat. no. 555329 and 555725; BD Pharmingen, respectively), in addition to 150 U/ml of interleukin 2 (Cat. no. PCYT-209; Eurobio). After differentiation/s or stimulation in the case of PBLs, cells were incubated with 1,000 U/ml of human IFNα2 before analysis (Cat. no. 11100–1; Tebu Bio).

### Ectopic DNA transfections, viral production, and confocal microscopy analyses

HEK293T cells were directly seeded on 0.01% poly-L-lysine-coated coverslips (Cat. no. P4832; Sigma-Aldrich) and analyzed 24 h after ectopic DNA transfection (unless otherwise specified, Lipofectamine 3000 Cat. no. L3000008; Thermo Fisher Scientific, according to the manufacturer's instructions). Cells were washed three times with PBS 1×, fixed with 4% paraformaldehyde (Cat. no. 15713; Euromedex) for 10 min, quenched with 50 mM $NH_4Cl$ (Cat. no. A4514; Sigma-Aldrich) for 10 min, and permeabilized with PBS–0.5% Triton X-100 (Cat. no. X100; Sigma-Aldrich) for 5 min. After a blocking step in PBS–5% milk, cells were incubated with primary antibodies for 1 h at room temperature (dilution 1:100), washed, and then incubated with fluorescent secondary antibodies (dilution 1:100). A 4'-5-diamidina-2-phenylindole (DAPI)-containing mounting medium was used (Cat. no. 62248; Thermo Fisher Scientific). Images were acquired using a spectral Zeiss LSM800 confocal microscope and analyzed with Fiji software (version 2.0.0). For 3D-reconstruction, z-stack collections were analyzed with the Imaris 9.2.0 software (Oxford Instruments Group). Colocalizations were quantified using the Pearson overlap coefficient (Fiji software).

IFITMs and PRRT2 coding DNAs were routinely transfected at a concentration of 1 µg per well of a 24-well plate along with 0.2 µg of VSV-G or of mEm.-VSV-G. When specified, increasing doses of GS15 coding DNA were also added (0, 0.2, 0.5, and 1 µg).

Monensin was used at a final concentration of 0.2 mM for 3 h, before analysis (Cat. no. M5273; Sigma-Aldrich), whereas when indicated, Amphotericin B (AmphoB, Cat. no. A9528; Sigma-Aldrich) was used for 18 h before confocal microscopy analysis at 1 µM.

Production of HIV-1 virion particles was instead carried out by calcium phosphate DNA transfection of the HIV-1 proviral clone NL4-3 (ratio 3 to 1, NL4-3 over IFITM). Virion particles released in the supernatant of transfected cells were harvested 48 h post transfection, syringe-filtered (0.45 µm filters, Cat. no. 146622; Minisart) to remove cellular debris and purified by ultracentrifugation over a 25% sucrose cushion (28,000 rpm for 2 h; SW41Ti rotor; Beckman Coulter ultracentrifuge).

In confocal microscopy experiments using the GM130 Golgi marker, IFITM or VSV-G relocalization was quantified for each cell as the ratio between the signal present at the Golgi over the signal present in the cytoplasm (Golgi/cytoplasm ratio). When the GM130 marker was not used, perinuclear accumulation was either scored visually (as in Fig 1, in light of the extremely clear phenotype; binary scoring: perinuclear or non-perinuclear), or quantified by determining the average percentage of accumulation of VSV-G at the left or right part of the nucleus (asymmetric distribution = perinuclear accumulation; symmetric distribution = non-perinuclear accumulation). To bypass artificial measurements due to differences in size between left and right portions of the cell, the average of the signals measured at each side of the cell were used. This measurement diminishes the absolute values of the protein present at one side, but offers the advantage of making measures independent from cell size differences. In light of the magnitude of the phenotype measured, this measurement yielded similar results than the binary visual scoring of the distribution of VSV-G or of IFITMs (perinuclear; non-perinuclear), so that one or the other was used, as indicated.

### Flow cytometry analysis

HeLa cells were transiently transfected with a VSV-G or a GFP coding vector by calcium-phosphate. 24 h later, the cells were detached

from the plate upon incubation with 10 mM EDTA in PBS, washed and labeled with a primary anti–VSV-G antibody (clone 41A1, kind gift of Frederick Arnaud, IVPC), followed by a secondary antibody (donkey anti-mouse IgG–Alexa Fluor 488 conjugate, Cat. no. A-21202; Life Technologies) to label plasma membrane VSV-G. Cells were then extensively washed, then fixed and analyzed by flow cytometry on a FACSCantoII (Becton Dickinson).

### Electron microscopy

Samples were fixed in a mixture of 4% paraformaldehyde (TAAB Laboratories Equipment Ltd, cat. F/P001) and 1% glutaraldehyde (Cat. no. 16310; Electron Microscopy Science) in 0.1 M phosphate buffer (pH 7.4) for 24 h, washed three × 30 min in 0.1 M of phosphate buffer, and post-fixed for 1 h with 2% osmium tetroxide (Cat. no. 19190; Electron Microscopy Science) in 0.15 M of phosphate buffer. After washing in 0.1 M of phosphate buffer for 20 min and two × 20 min in distillated $H_2O$, samples were dehydrated in a graded series of ethanol solutions (50% ethanol two × 10 min; 70% ethanol three × 15 min and last portion for 14 h; 90% ethanol three × 20 min; and 100% ethanol three × 20 min). Final dehydration was performed by 100% propylene oxide (PrOx; Thermo Fisher Scientific [Kandel] GmbH, Lot X19E013) three × 20 min. Then, samples were incubated in PrOx/EPON epoxy resin (Fluka) mixture in a 3:1 ratio for 2 h with closed caps, 16 h with open caps, and in 100% EPON for 24 h at room temperature. Samples were replaced in new 100% EPON and incubated at 37°C for 48 h and at 60°C for 48 h for polymerization. Serial ultra-thin sections (thickness 70 nm) were cut with a "Leica Ultracut UCT" ultramicrotome (Leica Microsysteme GmbH), placed on TEM nickel one-slot grids (Cat. no. G2500N; Agar Scientific, Ltd.) coated with Formvar film and stained 20 min with 5% uranyl acetate (Cat. no. 8473; Merck) and 5 min Reynolds lead citrate. The sections were then observed at 100 kV with a Jeol 1011 TEM (JEOL) connected to a Gatan digital camera driven by Digital Micrograph software (GMS 3; Gatan).

### Glycosidase treatment assays and WB quantification

Cell lysates expressing IFITM3 and VSV-G proteins were collected and split in three aliquots that were either untreated, or treated with Endoglycosidase H (EndoH, P0702S; NEB) or N-glycosilase F (PNGase F, P0704S; NEB), according to the manufacturer's instructions. Treated lysates were then analyzed by WB, images acquired using Image Lab Touch Software (version 2.0.0.27, Chemidoc Imaging System from Bio-Rad) and bands were quantified by densitometry using the volume tool in the same software.

### Software

Electron microscopy: Digital Micrograph software (GMS 3; Gatan). Confocal microscopy: Fiji software (version 2.0.0), Zen (version 2.3, Zeiss) and Imaris 9.2.0 software (Oxford Instruments Group). WB: Image Lab Touch Software (version 2.0.0.27, Chemidoc Imaging System from Bio-Rad). Flow cytometry: FlowJo (version X; BD). Statistics and graphs: GraphPad Prism8 (8.4.3; Graphpad software, LLC).

### Statistical analyses

The statistical analyses used in this study were calculated with the GraphPad Prism8 software: t tests (unpaired, two-tailed), one-way ANOVA tests with either Tukey's or Dunnett's multiple comparisons, as indicated in the figure legends.

### Ethics statement

Primary blood cells were obtained from the blood of healthy donors (EFS-Lyon) in the form of discarded "leukopacks" obtained anonymously so that gender, race, and age of donors are unknown to the investigator and inclusion of women, minorities or children cannot be determined. The work was carried out under the authorization no 19-606 from the Ethical Commission of the INSERM (CEEI) and no DC-2019-3774 CODECOH from the French Ministry of Research.

## Data Availability

Source data are provided with this article. There is no restriction on data availability.

## Supplementary Information

## Acknowledgements

Thanks to Delphine Muriaux for sharing insights, to Romain Appourchaux for his initial input, to Léa Picard and Lucie Etienne for help with bioinformatic analyses and evolutionary discussions. For sharing material used in this study, we are indebted to Howard Hang at Rockfeller University, New York, USA; Oleg Varlamov from the Oregon National Primate Research Center, USA; to Bryan Heit from the Western University, London, Canada; Frederick Arnaud from the IVPC, Lyon, France; as well as to the different authors of plasmids retrieved through Addgene cited in the Methods section. L Zhong and Y Song are the recipients of a PhD fellowship of the Chinese Scholarship Council; F Marziali is a post-doctoral fellow of the ANRS|Maladies infectieuses émergentes (ANRS|MIE). A Cimarelli is supported by the CNRS. Work in the laboratory of A Cimarelli is supported by grants from the ANRS (AO-2019-1 and AO-2021-1), as well as the ANR (ANR-20-CE15-0025-01). The funders had no role in study design, data collection and analysis, decision to publish, or preparation of the manuscript. We acknowledge the contribution of the microscopy (LYMIC-PLATIM) platform of SFR BioSciences Gerland Lyon Sud (UMS3444/US8).

### Author Contributions

L Zhong: formal analysis and investigation.
Y Song: formal analysis and investigation.
F Marziali: formal analysis and investigation.
R Uzbekov: data curation, formal analysis, and investigation.
X-N Nguyen: formal analysis and investigation.
C Journo: methodology.
P Roingeard: formal analysis and writing—original draft.

A Cimarelli: conceptualization, data curation, supervision, funding acquisition, writing—original draft, and project administration.

## Conflict of Interest Statement

The authors declare that they have no conflict of interest.

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
