## [Reviewer comments · Life Science Alliance]

Life Science Alliance

A novel domain within the CIL regulates egress of IFITM3 from the Golgi and reveals a regulatory role of IFITM3 on the secretory pathway

Li Zhong, Yuxin Song, Federico Marziali, Rustem Uzbekov, Xuan-Nhi Nguyen, Chloe Journo, Philippe Roingeard, and Andrea Cimorelli

DOI: <https://doi.org/10.26508/lsa.202101174>

Corresponding author(s): *Andrea Cimorelli, International Center for Infectiology Research*

Review Timeline:

Submission Date:	2021-07-26
Editorial Decision:	2021-07-28
Revision Received:	2022-02-15
Editorial Decision:	2022-03-07
Revision Received:	2022-03-21
Editorial Decision:	2022-03-22
Revision Received:	2022-03-24
Accepted:	2022-03-25

Scientific Editor: Novella Guidi

Transaction Report:

Please note that the manuscript was reviewed at Review Commons and these reports were taken into account in the decision-making process at Life Science Alliance.

July 28, 2021

Re: Life Science Alliance manuscript #LSA-2021-01174

Andrea CIMARELLI
International Center for Infectiology Research
46 Allée d'Italie
Lyon 69007
France

Dear Dr. Cimarelli,

Thank you for submitting your manuscript entitled "A novel domain within the CIL regulates egress of IFITM3 from the Golgi and prevents its deleterious accumulation in this apparatus" to Life Science Alliance. The manuscript was submitted and reviewed via Review Commons. The authors then chose to transfer their manuscript, along with the reviewers' comments and a proposed revised plan to Life Science Alliance (LSA). The reviewer comments and revision plan was assessed at LSA, and LSA editors deemed that the manuscript could be further considered at LSA provided the authors revise the manuscript, in accordance to what they have laid out in the pbp rebuttal / revision plan.

We, thus, encourage you to submit a revised manuscript to us that includes the previous underlined points by the LSA scientific editor:

- Address Reviewer 1's concern about the role of WT IFITM3 and whether it would have a role in maintaining Golgi function and trafficking with experiments using endogenous and exogenous WT IFITM3, and address all the minor points
- Address Reviewer 2's concerns regarding the FRET assay the authors used to analyse the vesicle fusion defect caused by the mutant IFITM3 and include additional controls to validate the assay. Also address all the minor points
- Address Reviewer 3's comments by providing more evidence that WT IFITM3 naturally trafficks through the Golgi and that the mutant form really inhibits vesicle fusion in the Golgi

Given that new data will be added to the revised manuscript, the revision will have to be looked at by a set of referees, most likely the same ones as Review Commons.

Thank you for this interesting contribution to Life Science Alliance. We are looking forward to receiving your revised manuscript.

Sincerely,

- A letter addressing the reviewers' comments point by point.
- An editable version of the final text (.DOC or .DOCX) is needed for copyediting (no PDFs).

B. MANUSCRIPT ORGANIZATION AND FORMATTING:

We appreciate the constructive feedback of the reviewers that improved our study. Please, find our answers (blue) to their specific concerns (black).

REVIEWER 1**GENERAL REMARKS**

The reviewer defines the findings generally interesting, but requiring more focus on the behavior of WT IFITM3 rather than mutant protein.

Accordingly, we have now included several experiments on WT IFITM3 (its distribution in different cells and the comparison of VSV-G trafficking in WT and IFITMs-KO cells) that support the notion that endogenous IFITMs do play a regulatory role in glycoprotein trafficking.

We are sure the reviewer would agree with us that the importance/functions of novel protein domains can only be ascertained through the characterization of the resulting mutants. In this respect, this mutant has been instrumental to reveal the importance that egress from the Golgi bears for the biology of IFITM3 and to focus on this novel regulatory function.

The reviewer mentions that trafficking disruptions are not novel and have been explained through the direct binding between IFITM3 and the C-ter tail of HIV-1 gp41 glycoprotein. We would like to point out that this interaction has not been confirmed by other labs neither for HIV, nor for other viral glycoproteins (that actually lack a similar corresponding portion).

MAJOR ISSUES

1) The reviewer suggests to extend our work to WT IFITM3.

Accordingly, Fig 4 is devoted to this topic. We provide evidence that: a) increased expression levels of WT IFITM3 lead to the protein progressive accumulation in the Golgi; b) this can be observed also with endogenous IFITM3 in different cells in which their expression is constitutive or IFN-inducible (HeLa; primary PBLs; primary macrophages and A549); c) Trafficking of VSV-G to the plasma membrane is more effective in IFITMs-KO than in WT HeLa cells, revealing that IFITMs do play a previously unrecognized role in the secretory pathway.

2) The reviewer asks us to more carefully re-examine the domain identified here among dispanins. They then argue that accumulation in the Golgi could be caused by the mutant protein misfolding, aggregation or degradation.

In agreement with their comments, we have restructured the paragraph and Fig 7, which are now more comprehensible. We do not believe that the 85-90 mutant is concentrated at the Golgi for reasons other than a trafficking defect. First, misfolded proteins and the ensuing unfolded protein response (UPR) should result in ER-retention which is not the case here. Second, we now provide evidence that AmphotericinB, compound known to relieve the

IFITM-driven membrane fusion defect, also relieves the phenotype of the 85-90 mutant, strongly suggesting that this mutant is fully competent for membrane fusion impairment.

3) The reviewer raises different issues: a) the possibility that the 85-90 mutant could increase lysosomal degradation of Env, as reported by Ahi et al; b) whether WT IFITM3 could also alter Golgi morphology and functions; c) whether the defect is Golgi-specific, given that Gag translation, which is not Golgi dependent, seems also affected.

a) While, in light of the drastic changes induced in the Golgi by the 85-90 IFITM3 mutant, changes in other compartments could occur, the quality but not the quantity of VSV-G is affected by the mutant, supporting the idea of a trafficking defect, rather than of a global increase in lysosomal functions; b) experiments displayed in Fig 4 indicate that WT IFITM3 also affects glycoprotein trafficking. Not surprisingly given that it possesses a functional egress domain, the phenotype of WT is milder than the one of the 85-90 mutant. c) While it is true that a decrease in Gag accumulation can be observed, it is of far lower magnitude than the one of Env. We better discuss this in the text (last paragraph, page 4).

4) The reviewer asks us to better describe the EndoH/PNGaseF sensitivity assay.

Accordingly, we have inserted a scheme and better explained the assay in the text (Fig 1e and second paragraph, page 5).

5) The reviewer asks us to determine whether trafficking defects are specific for viral glycoproteins or whether they can be observed also on cellular ones.

Accordingly, we have determined that such defects apply also to cellular glycoproteins by examining the behavior of CD93 (extensive data Fig 2)

4) Based on our discussion on the potential influence of the Golgi egress domain on the levels of palmitoylation at nearby cysteine residues, the reviewer asks us whether this could be directly tested by click chemistry.

We agree that exploring this would be of interest, but we have preferred to concentrate on the behavior of WT IFITM3, which was raised as the most important issue for our study.

MINOR ISSUES

1- Fig 3A on FRET experiments has been deleted, so this comment no longer applies.

2. We have replaced inner membranes for intracellular.

3. Although as the reviewer points out Gag-Pol is often used when referring to HIV-1, the correct nomenclature for *Retroviridae* (therefore including HIV) is Gag-Pro-Pol used here.

4. We have replaced decorticate for unravel.

SIGNIFICANCE and cross-comments

The main concern of the reviewer is to understand how much of the effects attributed to the IFITM3 mutants could be ascribed to *wild-type*.

We agree that this was a key point of our study and we have addressed this issue. We believe these results highlight a novel layer in the biology of IFITMs.

REVIEWER 2

MAJOR ISSUES

1) The Reviewer indicates that v- to t-SNARE fusion inhibition by IFITM3 is not supported by the results, because: a) GS15 is a *trans*-Golgi v-SNARE, while ERS24 is a *cis*-Golgi t-SNARE so that it is unclear where the two meet; b) that it is not sufficient for two proteins to reside in the same membrane to FRET and that molecular proximity/interaction is needed; c) that the study that developed the SNARE-tools used here did not use them for this purpose, indicating the need for additional validation controls; d) that the orientation of IFITM3 would be opposite with respect to the one described so far.

a) the reviewer is correct in stating that GS15 and ERS24 are *trans*- and *cis*-Golgi SNAREs, but these localizations represent a relative enrichment at these sites (for ex. PMID: 9211930; 7596416) with ample possibilities of overlap, that are likely increased upon overexpression. b-c) we have performed experiments with increasing doses of WT IFITM3 and also obtained a decrease in FRET. However, we have not been able to use NEM in cells overexpressing GS15 and ERS24 due to high toxicity. Given that this was a key control for the reviewer, we have decided to remove FRET experiments altogether, as it may weaken the study (new results with WT IFITM3 are appended for reviewer's only appreciation). As an alternative approach, we now provide evidence that AmphotericinB, compound known to relieve the membrane fusion defect of IFITMs, also relieves the defect of the 85-90 mutant (Fig 5a). In addition to the fact that GS15 v-SNARE overexpression (method described to relieve SNARE-dependent membrane fusion defects: PMID: 15102912; 12802061 etc) also relieves it, these results strongly point to membrane fusion defect in the Golgi. d) while the reviewer is correct about the expected orientation of the N-ter of IFITM3, we would like to point out that this portion can also be detected at the cell surface, suggesting a likely co-existence of the two conformations in cells (Tartour et al PPath-2017).

MINOR ISSUES

- 1) We have corrected the abstract and we confirm that the 85-90 mutant is the six-alanine mutant described in John, JVI-2013 and also in Appourchaux, JVI-2019).
- 2) We have commented on the reduction of Env and Gag (last paragraph, page 4). We ignore the reason for the slightly different migration in SDS-PAGE gel of the mutant protein.
- 3) We have marked that grey in the 3D-render of Fig 2b, indicates overlap between the red/green signals; we have corrected the scale bar and marked cis and trans in the Fig 2d. As the reviewer noted, the EM morphology of the Golgi is not textbook-like, but this is related to the cell type used, rather than to the procedure.
- 4) The cartoon in Fig 3A with the FRET experiment has been deleted.
- 5) Repetitive background information has been deleted.

6) Source data is provided.

SIGNIFICANCE and cross-comments

The reviewer describes this study as well-executed and thorough with findings of interest to virologists and cell biologists. They agree with the need to enlarge our study to the behavior of WT IFITM3 and to strengthen the FRET experiments

We thank the reviewer for these comments that we have addressed as mentioned above.

REVIEWER 3

GENERAL REMARKS AND MAJOR/MINOR ISSUES (NO DISTINCTION MADE BY THE REVIEWER)

The reviewer indicates the need for additional experiments supporting the conclusion that IFITM3 naturally transits through the Golgi.

We are unclear about this comment as transmembrane proteins, as IFITM3, are naturally synthesized via the ER-Golgi pathway. This contention is also well supported by our time-dependent analysis of IFITM3 distribution (Fig 3).

1) The reviewer asks us to explain why past studies failed to observe localization of IFITM3 at the Golgi and asks also to perform localization experiments with endogenous IFITM3.

On the whole, we believe not many studies examined this topic closely. While it is true that certain studies did not observe perinuclear accumulation of IFITM3, others did although specific Golgi markers were not examined (for ex. PMID: 28835547; 29503647; 30301809; 31212878). We now show that this is the case in a large selection of cells (Fig 4b).

2) The reviewer asks us why WT IFITM3 does not disrupt the Golgi functionalities.

We now provide evidence that WT IFITM3 also interferes with glycoprotein trafficking and accumulates at the Golgi. As expected, these phenotypes are less drastic than for the 85-90 mutant that is completely defective in Golgi egress. However, we believe more subtle phenotypes are even more interesting, because they suggest that IFITMs regulate Golgi trafficking in certain conditions, as for instance under IFN stimulation. We feel this represents a novel aspect of the biology of IFITMs and we hope it will stimulate new studies in the field.

3) The reviewer asks us whether the GM130-positive compartment is Golgi or a degradative compartment that co-stains with Lamp1 or LC3.

We believe the reviewer hypothesizes that Golgi retention is a consequence of protein misfolding and degradation, but we have not observed differences in the steady-state levels of VSV-G. Besides, misfolded proteins that transit from the ER-Golgi would be expected to be retained in the ER by the UPR and not in the *cis*-Golgi.

4) The reviewer says that Figure 4a (now Fig 3a) is missing a control with WT IFITM3

We are unclear about this comment as the Figure does present the analysis of the intracellular distributions of both WT and 85-90 IFITM3 over time.

5) The reviewer indicates that the fusion of two intracellular vesicles is not directly analogous to the fusion of a virus within an endosome with respect to the expected topology of SNAREs and IFITM3. They also comment that IFITM3 might be even expected to enhance fusion of intracellular vesicles, while perhaps preventing budding of new vesicles.

This point has been evoked also by Reviewer 2 (point 1d). In short, we believe the orientation of the Nter of IFITM3 is heterogeneous, so that intra and extra-cytoplasmic orientations can co-exist (Tartour et al PPath-2017) still able to mediate membrane fusion defects.

6) The reviewer indicates that it would be nice, though not essential, to show the distribution of the HIV-1 Env protein in the presence of the 85-90 IFITM3 mutant.

We now provide a qualitative analysis of HIV-1 Env that also accumulates in a perinuclearly (extended data Fig 1). We have not judged essential to carry it out quantitatively, as this would be redundant with the extensive analyses carried out with VSV-G.

SIGNIFICANCE and cross-comments

The reviewer defines the study as novel and important to shed light on the intracellular pattern of this important antiviral protein, but is concerned about the transposition to WT IFITM3. They also express concern about an effect on a degradative pathway.

We are pleased with these comments that we have addressed above

March 7, 2022

Re: Life Science Alliance manuscript #LSA-2021-01174R

Dr. Andrea CIMARELLI
International Center for Infectiology Research
46 Allée d'Italie
Lyon 69007
France

Dear Dr. CIMARELLI,

Thank you for submitting your revised manuscript entitled "A novel domain regulates egress of IFITM3 and prevents its deleterious accumulation in the Golgi" to Life Science Alliance. The manuscript has been seen by the original reviewers whose comments are appended below. While the reviewers continue to be overall positive about the work in terms of its suitability for Life Science Alliance, some important issues remain. We thus encourage you to address the remaining points raised by Reviewer 2.

Our general policy is that papers are considered through only one revision cycle; however, given that the suggested changes are relatively minor, we are open to one additional short round of revision. Please note that I will expect to make a final decision without additional reviewer input upon resubmission.

Please submit the final revision within one month, along with a letter that includes a point by point response to the remaining reviewer comments.

To upload the revised version of your manuscript, please log in to your account: <https://lsa.msubmit.net/cgi-bin/main.plex>
You will be guided to complete the submission of your revised manuscript and to fill in all necessary information.

B. MANUSCRIPT ORGANIZATION AND FORMATTING:

Sincerely,

Reviewer #1 (Comments to the Authors (Required)):

The authors have adequately addressed my previous concerns. Specifically, they have now shown convincingly that WT IFITM3

traffics through the Golgi apparatus, and that the effects of mutant IFITM3 on Golgi reorganization is due to its effects on membrane fusion as these effects were reversed by Amphotericin B and specific SNARE over expression.

My only comments is that I would strongly suggest that the authors reconsider the title as listed in their manuscript and return to a simpler title, such as what is listed in the manuscript submission system.

Reviewer #2 (Comments to the Authors (Required)):

In this revised manuscript, Zhong et al. present evidence for the existence of a sequence determinant allowing for egress of IFITM3 from the Golgi. The manuscript focuses mostly on the impact of a mutant that is lacking this determinant, which allows the authors to gauge the importance of IFITM3 on protein traffic (both viral and cellular) through the secretory pathway. The additional data added to this revision is appreciated and improves it. The most interesting finding is that amphotericin B can correct the Golgi defects caused by overexpression of mutant IFITM3, as can overexpression of GS15-CFP, which suggests that mutant IFITM3 is inhibiting some membrane fusion process to create this phenotype. However, these pieces of evidence need to be strengthened, in my opinion, with inclusion of IFITM3 WT. A limitation to this study is that it was previously shown that overexpressed and endogenous IFITM proteins inhibit viral glycoprotein trafficking through the secretory pathway (eg. it was shown that MLV Env accumulates at the plasma membrane to a greater extent in MEFs deficient for murine IFITMs). See below for major and minor comments.

Major:

1. Figure 5a: The effect of Amphotericin B on VSV-G localization in the Golgi should be performed for both IFITM3 WT and the 85-90 mutant, and the representative microscopy images should be shown for each condition. If needed, use increasing amounts of IFITM3 WT by transient transfection or induction with interferon. Currently, the summary bar graph depicting the perinuclear/non-perinuclear ratio of VSV-G does not correspond well to the images shown, which makes the data unconvincing. Again, in order to make this relevant to IFITM3 functions in vivo, a comparison with IFITM3 WT is required.
2. The title is poorly chosen if the authors want to emphasize what is important, interesting, and somewhat novel about this submission. It should mention that IFITM3 can inhibit trafficking of viral and cellular proteins through the Golgi, and this is why more evidence of how Amphotericin B/GS15-CFP negates this effect is needed.

Minor:

1. In their rebuttal and revision, the authors claim that they have provided extensive data for IFITM3 WT showing that it, too, interferes with Golgi trafficking of viral and cellular proteins subject to the secretory pathway. However, in the very first piece of data shown in Figure 1, IFITM3 WT has no effect on HIV Env incorporation into virions, despite it being overexpressed. Meanwhile, the 85-90 mutant clearly inhibits Env incorporation. This data, as presented, does not support the authors' message that restriction of viral glycoproteins in the Golgi is an activity performed by IFITM3 WT. Perhaps the authors should provide blot quantifications from multiple experiments to gauge whether WT inhibits Env incorporation (as been previously reported). While I believe that IFITM3 WT does exhibit the capacity to regulate Golgi-dependent trafficking (as evidenced by previous publications on IFITM function, which have been cited in this revision), the authors' decision to show this data first does not help the argument they are trying to make. The authors then fail to point out that IFITM3 WT actually does seem to promote HIV-1 Env accumulation in perinuclear sites when examined by IF microscopy, and the 85-90 mutant also promotes Env accumulation in perinuclear sites. The authors should better emphasize the impact that IFITM3 WT is having here to increase the appreciation of the work overall.
2. The authors do not provide any explanation for why elevated levels of IFITM3 WT (following transient transfection or interferon induction) do not progress through the Golgi normally. For example, how does interferon interfere with the motif that the authors identify to be important for Golgi egress?
3. Reference #1 is outdated in terms of what is known about IFITM mechanisms and relevance to several virus infections, including HIV.
4. The authors should have included IFITM3 WT when assessing the impact of IFITM3 on CD93 subcellular localization.
5. Remove "neutralizing" from "anti-VSV-G neutralizing antibody"

Referee cross-comments: The other reviewer seems satisfied with the changes that have been introduced into this revision. While the paper has been improved in terms of scientific soundness, the claims of novelty and significance are overstated.

Please, find below our answers (blue) to the remaining concerns of the reviewer (black).

REVIEWER 1

The reviewer suggests to consider a simpler/different title.

In agreement, we have provided what we feel is a more comprehensive title that reflects the key results of our study.

REVIEWER 2**GENERAL COMMENTS**

The reviewer asks us to determine whether AmphoB can also rescue the trafficking defects observed with high levels of WT IFITM3. They then question the novelty of our findings in light of the fact that a previous study reported increased plasma membrane accumulation of MLV-Env in IFITMs-KO-MEFs.

We have included not one, but two, sets of experiments that clearly indicate that AmphoB also relieves the defects caused by high levels of WT IFITM3, further strengthening our contention that WT IFITM3 plays a regulatory effect on the secretory pathway.

While it is true that increased plasma membrane accumulation of MLV-Env has been reported in IFITM-KO-MEF cells, this observation remains unexplained and anecdotal. In our study, we provide an explanation and a mechanism for it, by defining the domain that regulates IFITM3 trafficking through the Golgi and by defining the consequences that this novel step of the biology of IFITM3 bears for the secretory pathway. We feel these findings are novel.

MAJOR ISSUES

1) The reviewer asks us to determine whether AmphoB also relieves the trafficking defects caused by high levels of WT IFITM3.

Accordingly, we have performed AmphoB experiments in HEK293T cells with two different concentrations of WT IFITM3 (Fig 5a) and we have also examined the effects of this drug on VSVg trafficking in WT vs IFITMs-KO HeLa cells that express high levels of IFITMs (extended data Fig. 6b). These results clearly indicate first, that high levels of WT IFITM3 cause trafficking defects and second, that AmphoB relieves them.

2) The reviewer indicates that the pictures of Fig 5a (relating changes in the perinuclear distribution of VSVg in the presence/absence of AmphoB) do not reflect the graph.

We are unclear about this comment, but to better illustrate changes we now provide higher magnifications of single cells.

3) The reviewer indicates that the title should reflect more the fact that IFITM3 inhibits trafficking of viral and cellular proteins through the Golgi.

Accordingly, we have changed the title.

MINOR ISSUES

1) The reviewer indicates that in Fig. 1, WT IFITM3 fails to drive an Env incorporation defect into HIV virions, thus contradicting our contention.

We have better clarified the issue throughout the text. In the model we propose, the interference with the secretory pathway is the result of incomplete exit of IFITM3 from the Golgi. This can be observed: 1) through mutations in the domain responsible for this trafficking event, or 2) in the presence of high levels of WT IFITM3, that we believe saturate the Golgi egress pathway used by IFITM3. As such, we believe it is not surprising that low levels of WT IFITM3 exhibit negligible effects on HIV-1 Env .

2) Our working model is that high levels of expression of WT IFITM3 saturate the Golgi exit system leading to its clogging in this location. The exact molecular mechanism will be our next step.

3) We have not removed ref. 1, because we feel this review still provides interesting insights.

4) We have not performed experiments on CD93 and WT IFITM3, as we have preferred to concentrate on the experiments proposed by the reviewer on WT IFITM3 and AmphoB.

5) We have removed "neutralizing", as suggested

March 22, 2022

RE: Life Science Alliance Manuscript #LSA-2021-01174RR

Dr. Andrea CIMARELLI
International Center for Infectiology Research
46 Allée d'Italie
Lyon 69007
France

Dear Dr. CIMARELLI,

Thank you for submitting your revised manuscript entitled "A novel domain within the CIL regulates egress of IFITM3 from the Golgi and reveals a regulatory role of IFITM3 on the secretory pathway". We would be happy to publish your paper in Life Science Alliance pending final revisions necessary to meet our formatting guidelines.

- please upload your main and supplementary figures as single files
- please be sure that all Authors are added in the Author Contribution section in the manuscript file
- please use Capital letters when introducing panels in the figures, their legends, and callouts in the manuscript text
- LSA allows supplementary figures, but no EV Figures; please update your callouts for the Supplementary Figures in the manuscript Fig EV1A=Fig S1A; while supplementary figures use the system supplementary Fig S1, etc.
- Please indicate molecular weight next to each protein blot
- please revise the inset position in Figure 2B so that they match the zoomed-in parts
- please add a callout for Figure 6C to your main manuscript text

A. FINAL FILES:

B. MANUSCRIPT ORGANIZATION AND FORMATTING:

Sincerely,

March 25, 2022

RE: Life Science Alliance Manuscript #LSA-2021-01174RRR

Dr. Andrea Cimarelli
International Center for Infectiology Research
46 Allée d'Italie
Lyon 69007
France

Dear Dr. Cimarelli,

Thank you for submitting your Research Article entitled "A novel domain within the CIL regulates egress of IFITM3 from the Golgi and reveals a regulatory role of IFITM3 on the secretory pathway". It is a pleasure to let you know that your manuscript is now accepted for publication in Life Science Alliance. Congratulations on this interesting work.

DISTRIBUTION OF MATERIALS:

Again, congratulations on a very nice paper. I hope you found the review process to be constructive and are pleased with how the manuscript was handled editorially. We look forward to future exciting submissions from your lab.

Sincerely,
